# Formaldehyde Oxidation of Ce_0.8_Zr_0.2_O_2_ Nanocatalysts for Room Temperature: Kinetics and Effect of pH Value

**DOI:** 10.3390/nano13142074

**Published:** 2023-07-14

**Authors:** Zonglin Yang, Gaoyuan Qin, Ruijiu Tang, Lijuan Jia, Fang Wang, Tiancheng Liu

**Affiliations:** 1College of Chemistry and Environment, Yunnan Minzu University, Technology Innovation Team of Green Purification Technology for Industrial Waste Gas, Education Department of Yunnan, Key Laboratory of Environmental Functional Materials, Yunnan Province Education Department, Kunming 650504, China; yzong00692@163.com (Z.Y.);; 2China Energy Engineering Group Yunnan Electric Power Design Institute Co., Ltd., Kunming 650051, China; gyqin1053@ceec.net.cn

**Keywords:** formaldehyde, catalytic oxidation, Ce_0.8_Zr_o.2_O_2_ catalysts, kinetics parameters

## Abstract

Ce_0.8_Zr_0.2_O_2_ catalysts were prepared via the co-precipitation method under different pH conditions. The catalysts were characterized via TEM, XRD, XPS, BET, Raman, and FTIR. The oxidation performance of formaldehyde was tested. Precipitation pH affects the physicochemical properties and performance of the Ce_0.8_Zr_0.2_O_2_ catalyst. By controlling the precipitation pH at 10.5, the Ce_0.8_Zr_0.2_O_2_ catalyst with the largest specific surface area, the smallest grain size with the best formaldehyde removal rate (98.85%), abundant oxygen vacancies, and the best oxidation performance were obtained. Meanwhile, the kinetic parameters of the catalyst were experimentally investigated and the calculated activation energy was 12.6 kJ/mol and the number of reaction steps was 1.4 and 1.2.

## 1. Introduction

As a significant indoor contaminant, formaldehyde (HCHO) is primarily found in interior furniture and construction materials like wood, flooring, coatings, insulating materials, etc. [1]. Recent studies show that formaldehyde correlates significantly with human cancer and other dangerous diseases [2,3]. Therefore, eliminating indoor formaldehyde, especially at room temperature, has become an urgent and vital task.

Significant breakthroughs have been made in removing indoor formaldehyde pollution [4]. The main methods for removing formaldehyde are catalytic oxidation technology, adsorption technology, and photocatalysis technology. The catalytic oxidation approach has attracted much interest recently due to its high removal efficiency and cheap economic cost [5], and has been explored using a variety of substrate catalysts. The low-temperature catalytic oxidation method has the advantages of a wide application range and no secondary pollution [6], which has great research potential and application prospects. Some studies have found that composite oxide catalysts have an excellent effect on catalyzing formaldehyde oxidation at low temperatures [7]. As a rare earth element, cerium significantly impacts the removal of formaldehyde. CeO2 is widely used as an oxygen-storage material due to its fast and reversible redox cycle CeO_2_ ⇌ 1/2Ce_2_O_3_+1/2O_2_; simple CeO_2_ is prone to sintering at high temperatures and particle growth, leading to a reduction in specific surface area, thus reducing or losing oxygen-storage capacity. Lei Ma et al. [8] studied a new sodium-promoted Ag/CeO_2_ nanospherescatalyst, which could remove 90% of formaldehyde at low temperatures. CeO_2_-Co_3_O_4_ catalysts were produced by LU Suhong et al. [9] to obtain formaldehyde at 80 °C with a 100% removal rate. Yuanyuan Shi et al. [10] prepared a Pt -Ce/TiO_2_ catalyst that showed an excellent effect at room temperature. The catalysts mentioned above, CeO_2_, are sufficient to demonstrate Cerium’s higher performance. On the other hand, ZrO_2_ can have special catalytic effects when interacting with specific active components in the system because of its extensive surface oxygen vacancies and significant ion exchange capability [11]. Several previous papers have investigated the effect of precipitation pH on catalytic efficacy [12,13,14]. Clearly, pH is a critical parameter in the design of catalysts. However, no systematic attempts have been made to investigate the effect of pH on catalytic efficacy in the oxidative removal of formaldehyde from CZ catalysts. In addition, some studies have shown that kinetic reaction mechanisms and performance prediction models for catalysts can provide good evidence of better catalyst performance [15]. However, kinetic parameters and performance model predictions considering temperature and relevant component concentrations are rarely mentioned in most of the articles examining catalysts.

It is well known that the preparation conditions significantly affect the catalyst’s morphological structure and elemental valence state [16]. The effectiveness of cerium–zirconium solid solution on formaldehyde removal was determined via pre-experiments at different molar ratios, Ce_0.2_Zr0.8O_2_, Ce_0.4_Zr_0.6_O_2_, Ce_0.6_Zr_0.4_O_2_, and Ce_0.8_Zr_0.2_O_2_, with retention peaks of 75.32%, 81.43%, 86.5%, and 98.74%, respectively. The co-precipitation method was used to prepare a series of Ce_0.8_Zr_0.2_O_2_(CZ) catalysts with different precipitation pH. The efficiency of these samples for formaldehyde removal was investigated via SEM, XRD, XPS, Raman, FTIR, and other characterization methods. In addition, a kinetic study was carried out in this catalyst, which includes activation energy and reaction rate equation. The Langmuir–Hinshelwood model confirmed the catalytic mechanism and provided a basis for future room-temperature catalytic oxidation technology development.

## 2. Experiment

### 2.1. Preparation of Ce_x_Zr_1−x_O_2_ Catalyst

The catalyst was prepared using the co-precipitation method [17]. Zirconium nitrate pentahydrate (Macklin, Shanghai, China) and cerium nitrate hexahydrate (Macklin, Shanghai, China) were employed to prepare the catalysts. The specific procedure was as follows: in total, 0.8584 g of Zr(NO_3_)_4_·5H_2_O and 3.4736 g of Ce(NO_3_)_3_·6H_2_O were added into a beaker along with 50 mL of deionized water and the mixture was stirred at low speed with a magnetic stirring bar until the substances were completely dissolved. The pH was adjusted to 9, 9.5, 10, and 10.5 with ammonia and the mixture was stirred for 30 min. The Ce_0.8_Zr_0.2_O_2_ catalysts with pH 9, 9.5, 10, and 10.5 were obtained by calcination at 500 °C for 4 h at a heating rate of 5 °C/min and were named CZ-9, CZ-9.5, CZ-10, and CZ-10.5.

### 2.2. Catalytic Tests

Herein, the mixed gases of formaldehyde (HCHO), oxygen (O_2_), and nitrogen (N_2_) were utilized to simulate the formaldehyde gas environment in the room, and N_2_ was used as the balance gas. The catalyst CZ and 50 g SiO_2_ were loaded into a quartz tube (L: 800 mm; φ: 39 mm) and filled with quartz wool for fixation. The total gas flow rate was controlled with mass flow controllers (Sevenstar, Beijing, China). The reactor was equipped with a tubular heater and temperature controller, which could maintain the temperature in the range of 20–1000 °C. The temperature in the reactor was controlled with a thermocouple (Zhonghua, Tianjin, China).

The phenol reagent spectrophotometry method was utilized to quantify the formaldehyde concentration according to the China National Standard Methods for determining formaldehyde in the air of public places (GB/T18204.26.2000). 

The formaldehyde conversion (X_HCHO_) is calculated using the following equation:(1)$$X_{HCHO}=\frac{C_{\mathrm{in}}-C_{\mathrm{out}}}{C_{\mathrm{in}}}$$
where X_HCHO_ is the formaldehyde conversion (%), C_in_ is the inlet concentration of formaldehyde, and C_out_ is the outlet concentration of formaldehyde (ppm). 

It has been confirmed in many studies that the catalytic oxidation equation of formaldehyde was [18]:(2)$$HCHO+O_{2}\to H_{2}O+CO_{2}$$

According to the Arrhenius formula [19], the constant rate k of the catalytic oxidation of formaldehyde could be expressed as:(3)$$k=Ae^{-\frac{Ea}{RT}}$$
where k is the reaction rate constant, A is the pre-exponential factor, R is the gas constant value of 8.314 ($Jmol^{-1}K^{-1}$), T(K) is the reaction temperature, and Ea (kJ·mol) is the reaction activation energy.

The reaction rate equation of this reaction could be expressed under the Power-Rate law [20]. It is a classic pattern to explain the kinetics of the reaction, which can be represented in this work.
(4)$$r_{HCHO}=k[HCHO{]}^{\alpha }\cdot [O_{2}{]}^{\beta }$$
where $r_{HCHO}$ (ppm g^−1^ s^−1^) is the reaction rate, k is the reaction rate constant, and [HCHO] and [O_2_] are formaldehyde concentration and oxygen concentration, respectively. α and β are the reaction orders of HCHO and O_2_.

Substitute (3) into (4) to obtain:(5)$$r_{HCHO}=Ae^{-\frac{Ea}{RT}}[HCHO{]}^{\alpha }[O_{2}{]}^{\beta }$$

Taking the logarithm of the left and right sides of the equation, we obtain the following:(6)$$\ln r=-\frac{Ea}{RT}\ln A+\alpha \ln[HCHO]+\beta \ln[O_{2}]$$

The X_HCHO_ at various temperatures is determined from this equation while holding the incoming formaldehyde and oxygen concentrations constant [21]:(7)$$r=\frac{XCQ}{V_{m}m}$$
where r is the reaction rate, X_HCHO_ is the formaldehyde conversion rate, C is the formaldehyde inlet concentration, Q is the total gas flow, m is the mass of the catalyst, and V_m_ is the gas molar volume of 22.4 L/mol.

## 3. Results and Discussion

### 3.1. Characterization of Catalysts

#### 3.1.1. Transmission Electron Microscopy and BET Analysis

The structural characterization and elemental analysis of the CZ-10.5 catalyst were investigated using transmission electron microscopy (TEM). As is shown in Figure 1, the CZ-10.5 catalysts are depicted in (a), (b), (c) and (d) at different magnifications, respectively. The CZ-10.5 catalyst exhibits 50 nm agglomerates, as demonstrated in Figure 1a,b. The catalysts showed uniform nanosphere morphology with a diameter of approximately 10–20 nm; in addition, the lattice fringes of 0.16, 0.18, 0.26, 0.31, and 0.32 nm can be assigned to the corresponding (111), (200), (220), (311), and (222) planes of CeO_2_, respectively. The regions of Ce, O aggregation found in the catalyst particle suggest that Ce, O elements are equally distributed in the CZ-10.5 catalyst, according to the EDS elemental mapping shown in Figure 1d. It is interesting to notice that the Zr content on the EDS essential map is minimal, as evidenced by the few agglomerated patches, but the XPS fundamental analysis verified the existence of Zr. 

The effect of precipitation pH on the specific surface area and pore volume of the Ce_0.8_Zr_0.2_O_2_ (CZ) catalyst is given in Table 1. This means that when the pH of the system is lower than 9.1, some of the Ce^3+^ in the system may not be precipitated completely, resulting in slight differences in the composition and structure, resulting in a slightly lower specific surface area. At pH > 9.1, as the pH of the system increases, the more OH^−^ bound to the precipitate, the more difficult it is to orient the precipitate, making it more difficult to form crystalline forms. It is noteworthy that the average pore size of the samples obtained at pH values between 9.5 and 10.5 is relatively large. This is probably due to the relatively large amount of OH^−^ ions bound to the system, resulting in a larger precipitate size and a relatively larger pore size when the system is dehydrated. From a catalytic reaction kinetic point of view, the large pores are conducive to diffusion and facilitate the reaction.

#### 3.1.2. XRD Analysis of Surface Composition

The XRD was used to confirm the formation of a mixed ceria–zirconium oxide phase in the CZ catalysts and to identify zirconium incorporated into the cerium phase to form a ceria–zirconium complex. The XRD patterns are presented in Figure 2; reflection of CeO_2_ (PDF#43-1002) was observed.

Eight reflections from the cubic fluorite phase of CeO_2_ were observed compared to a standard card of CeO_2_ (PDF#43-1002) in the XRD patterns of the CZ samples. The locations of these reflections for the CeO_2_ sample created in this work are in good agreement with the pure ceria oxide. 

For all samples, the main diffraction peaks centered at 28.5°, 33.1°, 47.5°, and 56.3° were attributed to the corresponding planes (111), (200), (220), and (311) of cubic CeO_2_, respectively. At the same time, no ZrO_2_ signals can be observed because of the formation of a CeO_2_-ZrO_2_ solid solution, which is in accordance with the literature [22,23,24]. In addition, the introduction of Zr into CeO_2_ has been shown in several articles to enhance the lattice O mobility of CeO_2_, leading to cationic defects [25]. Slight shifts occur in the positions of these peaks corresponding to CeO_2_ (111) or the ZrO_2_ lattice plane, indicating intense interaction between ZrO_2_ and CeO_2_. The crystallite sizes calculated via the Scherrer equation are shown in Table 1, where the crystallite sizes are in the order: CZ-10 (10.7 ± 0.5 nm) > CZ-9 (10.2 ± 0.5 nm) > CZ-9.5 (9.9 ± 0.4 nm) > CZ-10.5 (8.2 ± 0.4 nm). The grain size of the samples decreased as the pH increased, and the size of the grains depended on the competition between the growth rate and the nucleation rate of the grains. An increase in pH means that the amount of OH^−^ increases during the reaction, resulting in a decrease in the concentration of the metal hydroxide solute in the solution and slower crystal growth. The CZ-10.5 sample had the smallest grain size—the smaller the size, the larger the surface-to-volume ratio—thus exposing more active sites, and thus also indicating the best formaldehyde removal [26].

For all samples, the main diffraction peaks centered at 28.5°, 33.1°, 47.5°, and 56.3° were attributed to the corresponding planes (111), (200), (220), and (311) of cubic CeO_2_, respectively. At the same time, no ZrO_2_ signals can be observed because of the formation of a CeO_2_-ZrO_2_ solid solution, which is in accordance with the literature [22,23,24]. In addition, the introduction of Zr into CeO_2_ has been shown in several articles to enhance the lattice O mobility of CeO_2_, leading to cationic defects [25]. Slight shifts occur in the positions of these peaks corresponding to CeO_2_ (111) or the ZrO_2_ lattice plane, indicating intense interaction between ZrO_2_ and CeO_2_. The crystallite sizes calculated via the Scherrer equation are shown in Table 1, where the crystallite sizes are in the order: CZ-10 (10.7 ± 0.5 nm) > CZ-9 (10.2 ± 0.5 nm) > CZ-9.5 (9.9 ± 0.4 nm) > CZ-10.5 (8.2 ± 0.4 nm). The grain size of the samples decreased as the pH increased, and the size of the grains depended on the competition between the growth rate and the nucleation rate of the grains. An increase in pH means that the amount of OH^−^ increases during the reaction, resulting in a decrease in the concentration of the metal hydroxide solute in the solution and slower crystal growth. The CZ-10.5 sample had the smallest grain size—the smaller the size, the larger the surface-to-volume ratio—thus exposing more active sites, and thus also indicating the best formaldehyde removal [26].

The calculated lattice parameters are in the order CZ-10.5 (5.41) > CZ-10 (5.39), CZ-9.5 (5.39) > CZ-9 (5.38), CeO_2_ (0.54), and the obtained values indicate that the lattice parameters of the samples are reduced as compared to the pure cerium oxide, which may be due to the localized symmetry distortions caused by the doping of Zr crystal defects, whereas the lattice parameter of all the samples is larger than that of zirconium oxide (0.51), which may be attributed to the incorporation of zirconium elements leading to a larger lattice parameter, which further indicates the formation of cerium–zirconium solid solution. The increase in lattice parameter with increasing pH may be due to the fact that the increase in pH helps Zr^4+^ enter the CeO_2_ lattice and produce lattice defects as well as more oxygen vacancies, which is also confirmed via XPS. This means that the unit cell of the cubic phase shrinks in this order. Confirming that smaller Zr^4+^ (ionic radius is 0.084 nm for 8-fold coordination) partially replaces Ce^4+^ (ionic radius is 0.097 nm) ones in the crystal lattice [27]. Thus, the conclusion that an alternative solid solution of cerium–zirconium oxide was formed is further confirmed. The absence of zirconium species diffraction peaks in the XRD spectra, despite Zr^4+^ in the enriched region being detected in the TEM-EDS, can be attributed to the relatively small amount of zirconium ions added during the preparation of the CZ catalyst and their incorporation into the CeO_2_ lattice. The decrease in XRD lattice parameters further supports this assertion. As can be seen from Table 1, the specific surface area and pore volume of the CZ catalysts were much larger than those of ZrO_2_, which indicates that the addition of Ce and its influence on the specific surface area and pore size of the material was greatly affected, while the specific surface area of the CZ catalysts was larger than that of CeO_2_, presumably because the incorporation of Zr^4+^ enhanced the thermal stability of the catalysts and made it easier to maintain a high specific surface area.

In conclusion, the increased pH slightly increased the CZ catalysts’ lattice parameters, and the Zr ions’ replacement of the Ce ions caused a lattice distortion that enhanced oxygen migration efficiency and improved oxidation performance.

The IR spectra of the CZ-10.5 sample before and after the reaction are given in Figure 3. It can be seen that there are absorption peaks at 3480 cm-1, and 1620 cm-1, which should come from the deformation of -OH in the water and the water molecules adsorbed on the surface, respectively [28,29]. In addition, the absorption peak near 1509 cm-1 should be a symmetric stretching of the surface hydroxyl group. The absorption peak at 1380 cm-1 is associated with the absorption peak of the Ce-O bond [30,31]. There is no absorption peak for the Zr-O bond in FTIR, which is consistent with the XRD results. Comparing the FTIR spectra of CZ-10.5 before and after the formaldehyde removal reaction, it was found that the functional groups of CZ-10.5 before and after the formaldehyde removal were basically the same as the original sample, which indicates that CeO_2_-ZrO_2_ has excellent durability or stability.

#### 3.1.3. XPS Results

In order to determine the atomic concentrations of the catalyst surface and the valence states of the components (Ce, Zr, and O) in the catalysts, XPS analyses of the composition and chemical form of the surface were performed on the CZ-9, CZ-9.5, CZ-10, and CZ-10.5 samples. Figure 4b shows that the structure of the Ce3d spectra is consistent with the presence of both Ce^3+^ and Ce^4+^ [27]. The label v is used for 3d 5/2 spin–orbit combinations, and the label u is used for 3d 3/2 spin–orbit combinations. Ce^4+^ consists of three double peaks, and Ce^3+^ consists of two double peaks. The structures labeled v, v″, v‴, u, u″, and u‴ are characteristic peaks of Ce^4+^, and the structures labeled v0, v′, u0, and u″ are characteristic peaks of Ce^3+^ [32,33,34]. Generally, the Ce^3+^/Ce^4+^ ratio is related to the amount of oxygen vacancies on the catalyst surface [35]. The ratio of Ce^3+^/Ce^4+^ is displayed in Table 1 as a consequence of the quantitative integral calculation results. Of all the catalysts, CZ-10.5 showed the most outstanding Ce^3+^/Ce^4+^ ratio (0.44) and a substantially greater Ce^3+^ concentration than the other samples. It is evident that a higher pH value causes the Ce^3+^/Ce^4+^ ratio to increase; there was also an increase in the Ce^3+^ specie. Additionally, it resulted in the Ce^3+^ feature migrating to the CZ-9, CZ-9.5, CZ-10, and CZ-10.5 crystals with higher binding energies. According to reports, larger Ce^3+^ concentrations result in more unsaturated chemical bonds on the catalyst surface and, as a result, more significant oxygen vacancy generation [36]. This suggests that CZ-10.5 has stronger formaldehyde catalytic performance than other samples.

The structure of the Zr 3d spectra of CZ catalysts is shown in Figure 4d. The shoulder peak, which is consistent with the Zr^4+^ species, is centered at 184.3 eV, while the main peak is at 182 eV [37]. Due to the inclusion of Ce^4+^ ions into the crystal structure of ZrO_2_, the reflections in the diffraction patterns of CZ samples are somewhat displaced to higher angles when compared to the diffraction pattern of zirconium oxide.

For what is shown in Figure 4c, sub-peaks of O 1 s were fitted to obtain detailed information on the surface chemical state of oxygen species. As shown in Figure 4c, the O 1 s binding energy for all CZ catalysts exhibited an expected main peak at 529.2 eV and a shoulder at 531 eV. They could be assigned to lattice oxygen (O_lat_) from CeO_2_ and surface chemisorbed oxygen (O_ads_), respectively [38,39]. The ratio of O_ads_/O_lat_ calculated in Table 1, obtained via XPS. The order was CZ-9.5 (0.93) > CZ-9 (0.89) > CZ-10.5 (0.86) > CZ-10 (0.44), suggesting that the different pH values could affect the number of oxygen species to exhibit a better catalytic performance of formaldehyde.

#### 3.1.4. Raman Analysis

Raman spectroscopy is very sensitive to crystal symmetry and helps provide additional structural information. The samples were studied via Raman spectroscopy to reveal their surface structure and phase composition. 

In Figure 5, the Raman spectra of CZ samples have a prominent peak located at 465 cm^−1^, which is attributed to the F_2g_ mode of the fluorite-type lattice; some studies have shown that the position of the main peak of CeO_2_ is located at 476 cm^−1^. In contrast, for all samples, the position of the main peaks is redshifted to varying degrees, which confirms the incorporation of zirconium into the CeO_2_ lattice in CZ [40]. The incorporation further influences the polarizability of the symmetrical stretching mode of the [Ce-O_8_] vibrational unit, leading to the shift from that in pure CeO_2_ (470 cm^−1^). 

For CZ-10.5, CZ-10 a peak centered at ~615 cm^−1^ was observed, attributed to the oxygen vacancies in the CeO_2_ lattice [40]. By the same determination condition, the relative concentration of oxygen vacancies in the CeO_2_-based mixed oxides can be represented by the intensity ratio of the bands at 470 cm^−1^ and 615 cm^−1^ [41,42]; the order of I_470_/I_615_ is CZ-10.5 (2.36) > CZ-10 (1.24) > CZ-9 (1.18) > CZ-9.5 (0.93). It could be concluded that the oxygen vacancies of CZ-10.5 are higher than those of other CZ samples, which could deliver a better performance.

### 3.2. Catalytic Activity in HCHO Oxidation at Room Temperature

Figure 6 shows the formaldehyde oxidation performance of all prepared CZ catalyst samples at room temperature (298 K), where the catalyst activity is evaluated regarding the conversion of formaldehyde X_HCHO_. As shown in Figure 6, all samples exhibited good oxidation performance (>95%) within the first 1 h, which was attributed to the excellent oxidation performance of the ceria–zirconia oxides, especially the CZ-10.5 catalyst, achieving a removal rate of 98.85%. All samples showed a decrease in oxidation performance over the next hour of testing, which was attributed to a reduction in active sites due to the pore size of the catalyst being blocked by oxidation intermediates. In order to better investigate the oxidation efficacy of the CZ-10.5 catalyst at room temperature, the following kinetic experiments were conducted with the main objective of calculating the activation energy and kinetic equation of the CZ-10.5 catalyst.

Before conducting kinetic tests, experiments on the catalyst’s internal and external diffusion elimination are required. The catalyst concentration was regulated in several investigations, and the size was altered to prevent internal and external diffusion. As the W/FHCHO was below 2 × 10^−2^ (g·s·mL^−1^) and the catalyst particle size was 80–100 mesh under the condition (the feed gas of 20 vol% HCHO, 20 vol% O_2_, 55 vol% N_2_ balance with or without 5 vol% H_2_O, and the GHSV of 60,000 h^−1^, 298 K), we utilized a similar method to establish the criteria for removing internal and external diffusion in this case, confirming that internal and external diffusion could be eliminated [21,43]. As a result, the system can be considered a differential reaction system, with the surface reaction step acting as the rate-controlling step, allowing for studying the formaldehyde catalytic oxidation kinetics.

The Arrhenius plots for HCHO oxidation are shown in Figure 7. The activation energy date can be calculated with Table 2, and the catalytic oxidation of formaldehyde on the CZ-10.5 catalyst has an activation energy of 12.6 kJ/mol; these calculations proved to be some of the markers for judging the catalyst’s efficiency [44]. The CZ-10.5 catalyst has a low activation energy (12.6 kJ/mol), indicating that its oxidation efficiency is insensitive to temperature fluctuations and may operate at low temperatures while still achieving 98.85% oxidation efficiency (298 K). The HCHO conversion rate was discovered when the influence of formaldehyde concentration was investigated while maintaining the various atmospheric conditions at [HCHO] concentrations of 50–65 (ppm), under 298 K, 323 K, and 373 K, respectively. In Figure 8, the calculated response rate is displayed. Table 3 shows the slope value, which functioned as a stand-in for the sequence of reactions.

#### 3.2.1. Effect of Temperature and Concentration

The formaldehyde concentration should be maintained constant at room temperature with an O_2_ volume percentage of 15–30% while the impact of O_2_ concentration is studied. After measuring and computing the HCHO conversion rate, the determined response rate is shown in Figure 9. The slope value, which indicates the order of responses, is displayed in Table 4.

According to the information provided above, the rate equation for the formaldehyde catalytic oxidation reaction at room temperature could be represented as follows:(8)$$r_{HCHO}=203.5\exp(\frac{-8.4092}{RT}){\left[HCHO\right]}^{1.4}{\left[O_{2}\right]}^{1.2}$$

#### 3.2.2. The Langmuir–Hinshelwood Kinetic Model

Multiple variables are frequently present in the kinetic model of a catalytic process. In this experiment, the experimental data were fitted with the Langmuir–Hinshelwood mechanism to assess the catalytic reaction mechanism while considering the various reaction pathways. The adsorption of HCHO and O_2_ on the catalyst surface and the desorption of H_2_O and CO_2_ from the catalyst are the two primary processes in the oxidation of formaldehyde over CZ catalysts. The following sentence describes how formaldehyde reacts with the catalyst:(9)$$HCHO+*\overset{K_{HCHO}}{↔}HCHO*\;;\;O_{2}+*\overset{K_{O_{2}}}{↔}O*;\;HCHO*+O*\overset{k}{↔}CO_{2}*+H_{2}O*;\;CO_{2}*\overset{K_{CO_{2}}}{↔}CO_{2}+*;\;H_{2}O*\overset{K_{H_{2}O}}{↔}H_{2}+*$$

K_HCHO_, K_O_2__, K_CO_2__, and K_H2O_ are the adsorption equilibrium constants for the corresponding compound, and k is the reaction constant.

The adsorbing HCHO and O_2_ reaction was regarded as a rate-determination step through the above reaction process. The reaction could be expressed as:(10)r=kθθHCHOO2

According to the Langmuir–Hinshelwood mechanism $\theta_{HCHO}$, $\theta_{O_{2}}$, it could be expressed as:(11)$$\theta_{HCHO}=\frac{K_{HCHO}C_{HCHO}}{1+K_{HCHO}C_{HCHO}+K_{O_{2}}C_{O_{2}}+K_{H_{2}O}C_{H_{2}O}+K_{CO_{2}}C_{CO_{2}}}$$
(12)$$\theta_{O_{2}}=\frac{K_{O_{2}}C_{O_{2}}}{1+K_{HCHO}C_{HCHO}+K_{O_{2}}C_{O_{2}}+K_{H_{2}O}C_{H_{2}O}+K_{CO_{2}}C_{CO_{2}}}$$

Substitute (10) and (11) into (9) to obtain:(13)$$r=\frac{kK_{HCHO}C_{HCHO}K_{O_{2}}C_{O_{2}}}{(1+K_{HCHO}C_{HCHO}+K_{O_{2}}C_{O_{2}}+K_{H_{2}O}C_{H_{2}O}+K_{CO_{2}}C_{CO_{2}}{)}^{2}}$$

According to the experiment’s circumstances, CO_2_ and H_2_O remained constant, and the oxygen concentration outweighed the actual oxygen requirement. Consequently, (12) might be expressed simply as:(14)$$r=\frac{k^{\prime }K_{HCHO}^{\prime }C_{HCHO}}{(1+K_{HCHO}^{\prime }C_{HCHO}{)}^{2}}$$
where $k^{\prime }$ is the apparent reaction rate constant, $K^{\prime }$ is the apparent adsorption constant.

Figure 10 displays the experimental data and the fitting curve. Table 5 lists the relevant empirical results. According to how the L-H model was interpreted, formaldehyde molecules and oxygen molecules were simultaneously adsorbed on the active site before moving on to the next step. The appropriate correlation coefficient was 0.9683, as shown in Table 5, sufficient to support the catalytic oxidation mechanism. It has been demonstrated from the studies above that the L-H mechanism is responsible for the catalytic oxidation of formaldehyde on the CZ catalyst. According to the findings of earlier studies and the L-H model, the catalytic oxidation mechanism of formaldehyde on Ce_x_Zr_1−x_O_y_ catalysts is depicted in Figure 11. Formaldehyde catalytic oxidation is the procedure’s name. Surface oxygen species such as surface-adsorbed oxygen and surface hydroxyl oxygen are produced when oxygen first interacts with the catalytic surface. Active sites and surface oxygen species aided the transformation of formaldehyde gas on the metal surface into carbon dioxide and water. Ce^4+^ was converted throughout this process into Ce^3+^. Due to the shared anion with Zr^4+^, Ce^3+^ was converted back into Ce^4+^ to maintain activity and continue the reaction. This procedure was consistent with earlier studies’ descriptions of the L-H model [45].

## 4. Conclusions

In summary, a series of Ce_0.8_Zr_0.2_O nanocatalysts (prepared at different pH values: 9, 9.5, 10, 10.5) for the catalytic oxidation of formaldehyde were synthesized. The prepared CZ catalysts showed good oxidation activity at room temperature and maintained a more than 70% catalytic efficiency within 6 h. In particular, the CZ-10.5 catalyst has a more extended catalytic performance due to its larger specific surface area and more oxygen vacancies than the other catalysts. TEM results show that Ce and O elements are the main elements in the CZ-10.5 catalyst, while Zr elements show smaller agglomerates. XRD and XPS results show that introducing a moderate amount of Zr elements improves the catalyst lattice and significantly increases the Ce^3+^ concentration. It substantially increases the amount of Ce^3+^ to enhance the number of Ce^3+^/Ce^4+^ oxidation couples, promotes the redox cycle of Ce^3+^/Ce^4+^, generates more oxygen vacancies, and increases the oxygen migration rate, including surface-adsorbed oxygen and lattice oxygen species. In subsequent kinetic experiments, it was concluded that the lower activation energy made the CZ catalyst insensitive to changes in temperature, resulting in a strong oxidation performance at low temperatures, and the results of the fit to the L-H model indicated a catalytic oxidation mechanism consistent with that of the CZ catalyst.

## Figures and Tables

**Figure 1 nanomaterials-13-02074-f001:**
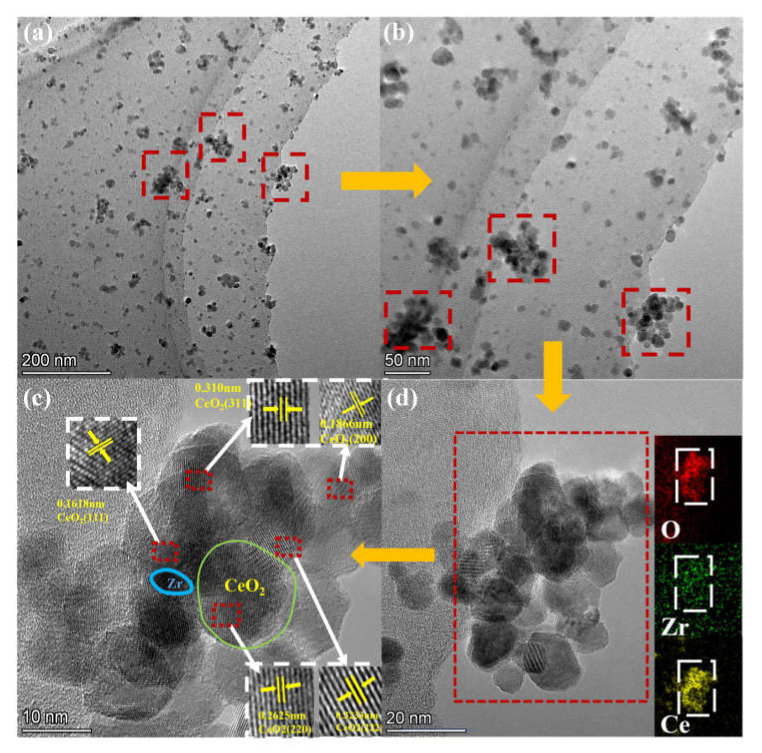
TEM images and corresponding elemental mapping of CZ-10.5 catalyst, (**a**) 200 nm; (**b**) 50 nm; (**c**) 10 nm; (**d**) 20 nm.

**Figure 2 nanomaterials-13-02074-f002:**
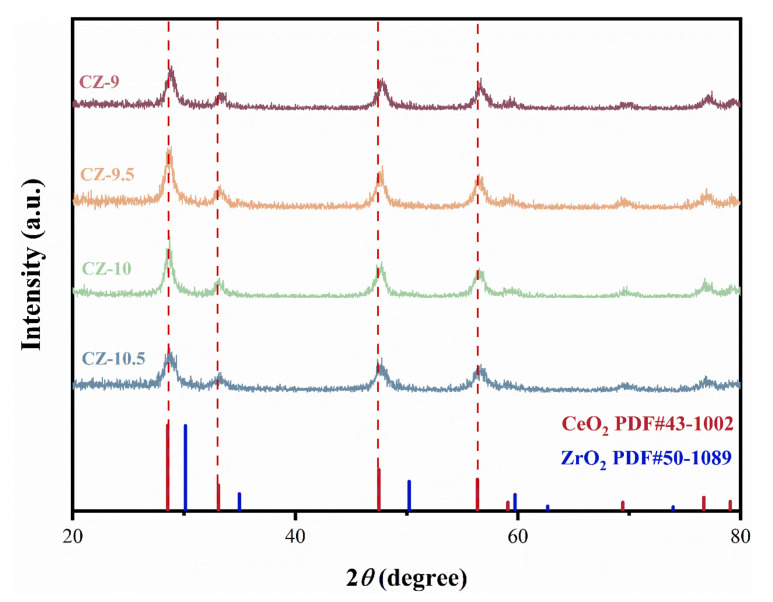
XRD patterns of CZ catalysts prepared at different pH values.

**Figure 3 nanomaterials-13-02074-f003:**
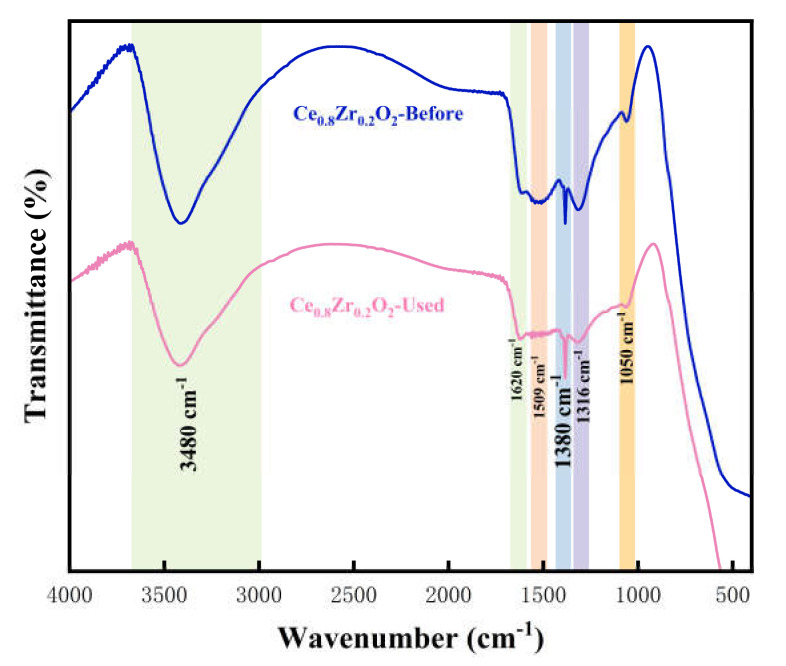
Infrared spectra of CZ-10.5 before and after reaction.

**Figure 4 nanomaterials-13-02074-f004:**
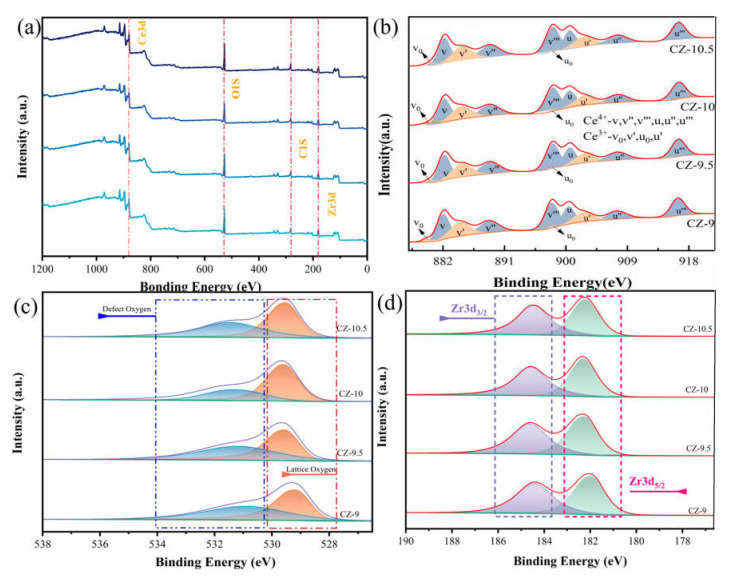
XPS spectra of the prepared samples: (**a**) XPS measured spectra, (**b**) Ce 3d, (**c**) Zr 3d, and (**d**) O 1 s.

**Figure 5 nanomaterials-13-02074-f005:**
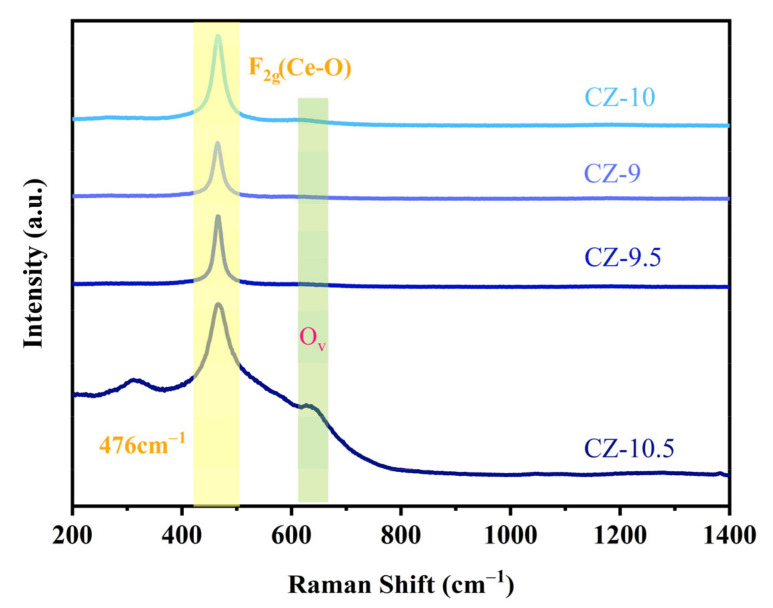
Raman spectra of CZ catalysts prepared at different pH values.

**Figure 6 nanomaterials-13-02074-f006:**
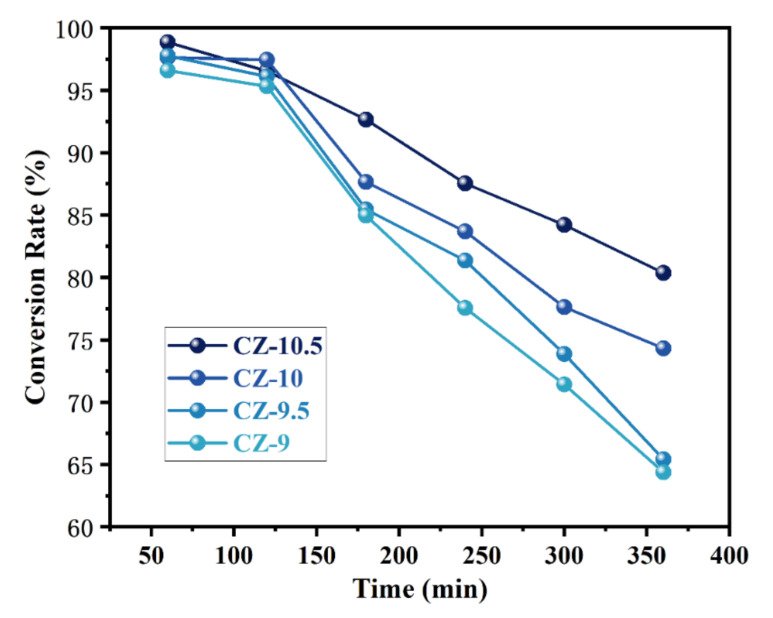
HCHO conversion rate of CZ catalysts prepared at different pH values (the feed gas of 20 vol% HCHO, 20 vol% O_2_ and 55 vol% N_2_ balance with or without 5 vol% H_2_O and the GHSV of 60,000 mL·g^−1^·h^−1^, 298 K).

**Figure 7 nanomaterials-13-02074-f007:**
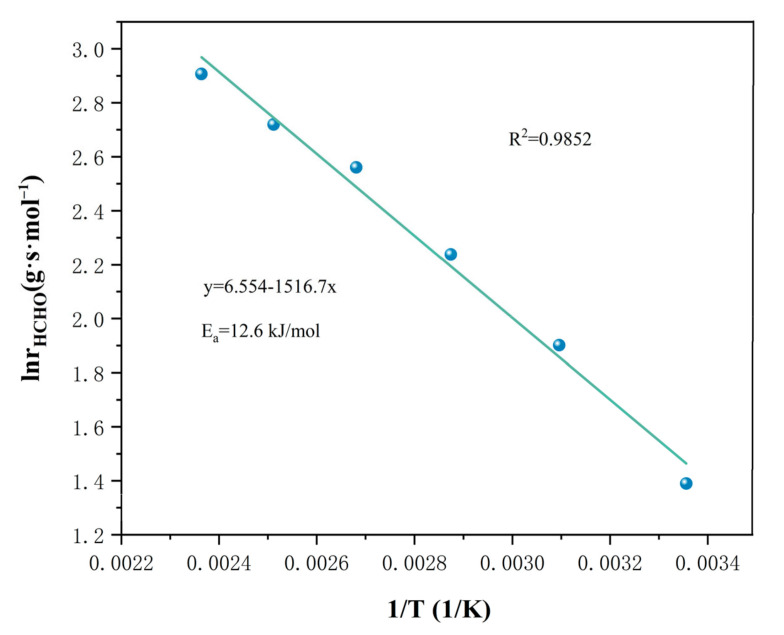
Arrhenius plots for HCHO oxidation.

**Figure 8 nanomaterials-13-02074-f008:**
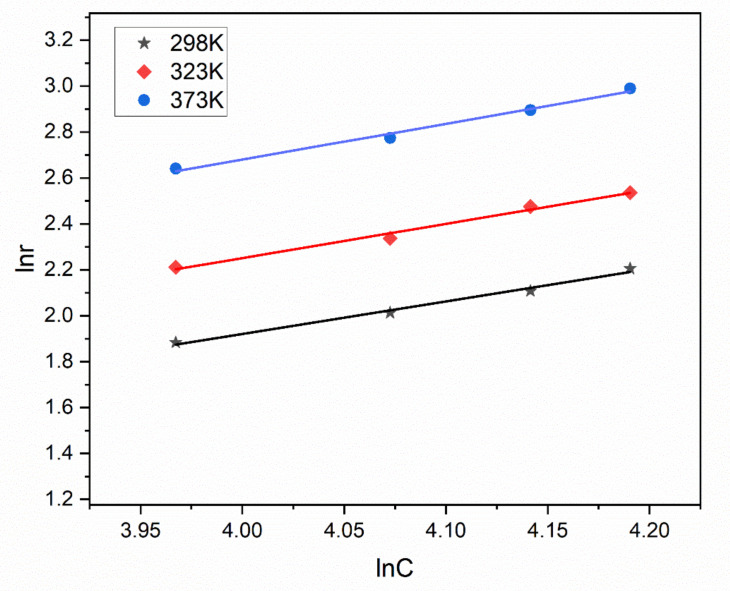
Linear regression of HCHO reaction order over CZ-10.5 catalysts.

**Figure 9 nanomaterials-13-02074-f009:**
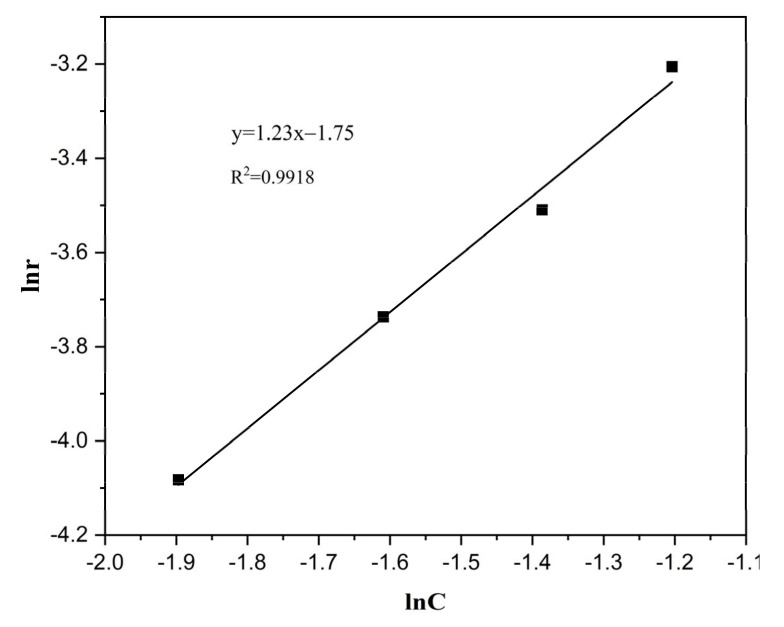
Linear regression of O_2_ reaction order over CZ-10.5 catalysts.

**Figure 10 nanomaterials-13-02074-f010:**
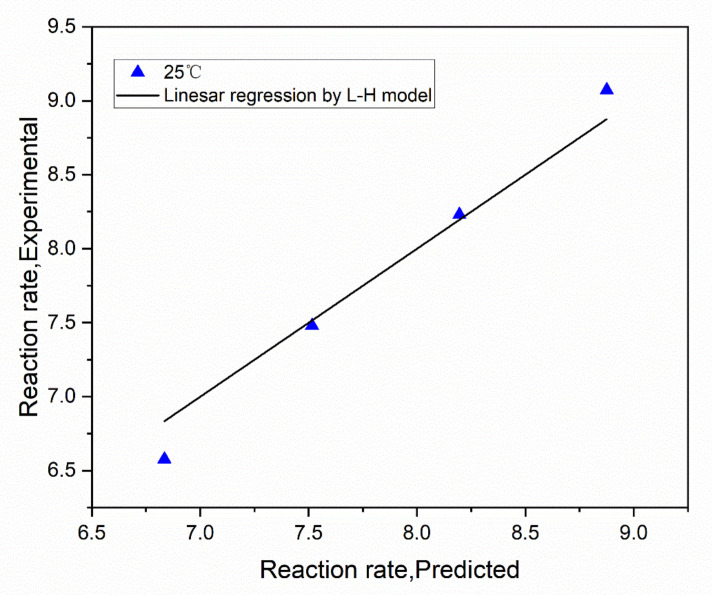
Parity plot comparing measured reaction rates with the predicted reaction rates with the single variable L-H model.

**Figure 11 nanomaterials-13-02074-f011:**
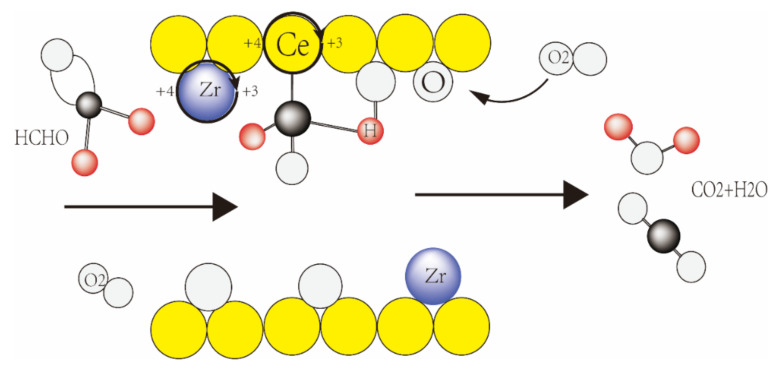
Catalytic Oxidation Mechanism of Formaldehyde over CexZr1−xOy Catalysts.

**Table 1 nanomaterials-13-02074-t001:** Ce, Zr, O content, S_BET_, total pore volume (V_pore_), average crystallite size, lattice parameter, XPS, and XRD results for the studied samples.

Sample	S_BET_ ^a^	Vpore ^b^	XRD Data	Average Crystallite	XPS Data		
Unit	m^2^/g	cm^3^·g^−1^	Lattice Parameter a, Å	Size, nm	Content, at.%		
					Surface Ce^3+^	Surface Ce^3+^/Ce^4+^	Surface O_ads_/O_lat_
CZ-10.5	84	0.101	5.409	8.2 ± 0.4	30.81	0.44	0.86
CZ-10	81	0.102	5.391	10.7 ± 0.5	25.16	0.34	0.44
CZ-9.5	81	0.104	5.389	9.9 ± 0.4	23.05	0.23	0.93
CZ-9	76	0.094	5.381	10.2 ± 0.5	24.70	0.33	0.89
CeO_2_	49	0.109	5.411	10.7	—	—	—
ZrO_2_	3	0.006	5.150	24.1	—	—	—

^a^ N_2_ isotherms at −196 °C were used to determine the specific surface areas through the BET equation (SBET). ^b^ Specific pore volume calculated at P/P_0_ = 0.98.

**Table 2 nanomaterials-13-02074-t002:** Ce_x_Zr_1−x_O_y_ catalyst reaction activation energy data.

Catalyst	Linear Regression Equation	R^2^	Ea (kJ/mol)
CZ-10.5	y = −1516.7x + 6.554	0.9852	12.6

**Table 3 nanomaterials-13-02074-t003:** Data of HCHO Reaction Orders over Ce_x_Zr_1−x_O_y_ Catalysts.

Temperature	Linear Regression Equation	R^2^	α
298	y = 1.41x − 3.73	0.9892	1.41
323	y = 1.49x − 3.71	0.9880	1.49
373	y = 1.56x − 3.54	0.9895	1.56

**Table 4 nanomaterials-13-02074-t004:** Data of O_2_ Reaction Orders over Ce_x_Zr_1−x_O_y_ Catalysts.

Linear Regression Equation	R^2^	β
Y = 1.23x − 1.75	0.9918	1.23

**Table 5 nanomaterials-13-02074-t005:** Date of Langmuir–Hinshelwood model.

Model	Rate Expression	k	K	R^2^
L-H	$r=\frac{k^{\prime }K_{HCHO}^{\prime }C_{HCHO}}{(1+K_{HCHO}^{\prime }C_{HCHO}{)}^{2}}$	3978	3.44 × 10^−5^	0.9683

## Data Availability

Due to the nature of this research, participants of this study did not agree for their data to be shared publicly, so supporting data is not available.

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
