# Peer review of "Formaldehyde Oxidation of Ce0.8Zr0.2O2 Nanocatalysts for Room Temperature: Kinetics and Effect of pH Value"

_nanomaterials, 2023, doi:10.3390/nano13142074_

Round 1

Reviewer 1 Report

After carefully reviewing the manuscript titled "Formaldehyde oxidation of Ce0.8Zr0.2O Nanocatalysts for room temperature: Kinetics and Effect of pH value," I recommend a major revision of the manuscript before considering it for possible publication in Nanomaterials. The study addresses the important topic of formaldehyde oxidation using Ce0.8Zr0.2O (CZ) nanocatalysts at room temperature. While the authors have conducted comprehensive experiments and provided valuable insights into catalytic efficiency, kinetics, and the effect of pH, several areas require significant attention and improvement. The characterization results need to be clarified. Detailed information on the composition and structure should be provided.

1. Consider reorganizing the abstract to follow a more logical structure, such as introducing the catalyst, describing the experimental characterization, presenting the pH effect, and concluding with the kinetic parameters.

2. There are some grammatical errors and inconsistencies in the manuscript that need to be addressed. Please proofread the manuscript and ensure that the language is clear, concise, and free of errors.

3. Elemental composition of the prepared materials should be studied quantitatively.

4. Introduction:

-The introduction provides a comprehensive overview of the significance of formaldehyde as an indoor pollutant and the need for effective removal methods. However, it could be further improved by explicitly stating the objective or research question addressed in this study.

-Consider providing more details and explanations regarding the findings of the mentioned studies, particularly regarding the catalytic performance and mechanism of CeO2 and ZrO2 catalysts.

5. XRD results:

- Clarify that the absence of ZrO2 signals in the XRD patterns is consistent with the formation of a CeO2-ZrO2 solid solution, as supported by previous literature references.

- Elaborate on the significance of the shrinking unit cell and the implications of Zr4+ partially replacing Ce4+ in the crystal lattice.

- Clarify the connection between the observed lattice parameter order and the formation of the alternative solid solution of cerium-zirconium oxide.

- The authors should provide further clarification regarding the formation of Ce0.8Zr0.2O in the prepared material. It is essential for the readers and the scientific community to have a clear understanding of the reasoning behind the proposed composition.

6. XPS:

- There were no significant differences observed in the XPS spectra between CZ-9, CZ-9.5, CZ-10, and CZ-10.5 samples. The authors should explore the implications of XPS findings for the catalytic performance of the CZ catalysts and compare them with other characterization techniques used in the study.

7. FTIR spectroscopy should be conducted to detect functional groups present in the CZ catalysts. Several studies have successfully utilized FTIR spectroscopy to identify and characterize functional groups in similar catalyst systems. The authors can consider citing the following papers as references:

Journal of Colloid and Interface Science 539:135-45. https://doi.org/10.1016/j.jcis.2018.12.052

Ultrasonics Sonochemistry 41:503-13. https://doi.org/10.1016/j.ultsonch.2017.10.013

Ultrason. Sonochem. 39 (2017) 540-549. https://doi.org/10.1016/j.ultsonch.2017.05.023

Extensive editing of English language required

Reviewer 2 Report

The paper entitled: Formaldehyde oxidation of Ce0.8Zr0.2O Nanocatalysts for room temperature: Kinetics and Effect of pH value, by: Z. Yang, G. Qin, R. Tang, L. Jia, F. Wang, and T. Liu, deals with the formaldehyde oxidation removal rate of Ce0.8Zr0.2O (CZ) catalysts at room temperature , the `paper describes the preparation of a series of CZ catalysts, that are described in terms of TEM, XRD, XPS, BET, and Raman for their characterization. One of the CZ catalyst exhibited an optimum oxidation efficiency at controlled pH, and the authors give a plausible mechanism for the activity. The authors describe the intrinsic kinetic parameters of the catalyst obtained by experiments and calculate the activation energy and the reaction orders. The paper was well described an suitable for publication as it is.

English is fine

Reviewer 3 Report

Ce0.8Zr0.2O the chemical formula is  not written. It is better to use Ce0.8Zr0.2O2 or Ce0.8Zr0.2O2-x.

From the text, the prerequisites for the choice of Ce0.8Zr0.2O  composition are not clear

Table 1, authors should add correct the definition error of crystallite size and review the values.

According to fig 2, XRD patterns of samples look similar, also the shift of position of XRD peaks is not observed. However, Table 1 contains values of lattice parameters , which are rather different. See, for example, 5.358 and 5.636A for  and CZ-10 catalyst. I highly recommended to recalculate these values and compare the obtained lattice parameters with ones for bare ceria and zirconia and literature data for CZ.

What effect does pH have on the precipitation process? There is a possibility that not all cations are precipitated. Authors should prove elemental composition of catalyst by independent method.

«It is not difficult to conclude that the effect of synthetic pH on the CZ  catalysts is reflected in the higher pH producing larger grain sizes.» – On the other hand, the possibility or change in the degree of incorporation of zirconium into cerium oxide cannot be ruled out. Please, comment.

«The lattice parameter  order derived from the calculations is CZ-10.5>CZ-10>CZ-9.5>CZ-9, indicating that higher  pH leads to smaller lattice parameters.» – contradicts the data in the table 1.

A decrease in pH leads to a decrease in the parameter, which is associated with an increase in the degree of substitution of cerium by zirconium cations, i.e. at pH 10-10.5, not all zirconium is in the composition of the solid solution. Therefore, for these preparation conditions and the specific surface area is higher.

Conclusions «XRD and XPS results show that introducing a moderate amount of Zr elements improves the catalyst lattice and significantly increases the Ce3+ concentration.» – Rather, on the contrary, the introduction of Zr into ceria leads to a decrease in Ce3+

Round 2

Reviewer 1 Report

The authors have addressed my comments and concerns in the revised version. Overall, the manuscript is well written and the reported results are of valuable interest to readers. I have no additional comments and I recommend accepting the paper.

Reviewer 3 Report

Table 1. Average crystallite size. It is nesessery to give errors for average crystallite size. Does 0.001 nm in 8.168 nm have a sence? also in lines 170-171

lines 179-181, "The lattice parameter order derived from the calculations is CZ-10.5(5.409)>CZ- 179 10(5.392)>CZ-9.5(5.389)>CZ-9(5.381), all samples have a lattice parameter smaller than 180 CeO2 (0.5411nm), and all samples have a lattice parameter greater than that of zirconia 181 (0.5125 nm), indicating that higher pH leads to greater lattice parameters" It is not clear how to compare 5.409 and 0.5211 nm
